# Synergistic Piezo-Photocatalysis of BiOCl/NaNbO_3_ Heterojunction Piezoelectric Composite for High-Efficient Organic Pollutant Degradation

**DOI:** 10.3390/nano12030353

**Published:** 2022-01-22

**Authors:** Li Li, Wenjun Cao, Jiahao Yao, Wei Liu, Feng Li, Chunchang Wang

**Affiliations:** 1Laboratory of Dielectric Functional Materials, School of Materials Science & Engineering, Anhui University, Hefei 230601, China; b20101003@stu.ahu.edu.cn (L.L.); caowenjun1206@126.com (W.C.); yjh1074457726@163.com (J.Y.); b20301114@stu.ahu.edu.cn (W.L.); 2Information Materials and Intelligent Sensing Laboratory of Anhui Province, Institutes of Physical Science and Information Technology, Anhui University, Hefei 230601, China

**Keywords:** BiOCl/NaNbO_3_, heterojunction, piezocatalysis, photocatalysis, degradation

## Abstract

Piezo-photocatalytic technique is a new-emerging strategy to alleviate photoinduced charge recombination and thus enhance catalytic performance. The heterojunction construction engineering is a powerful approach to improve photocatalytic performance. Herein, the BiOCl/NaNbO_3_ with different molar ratios piezoelectric composites were successfully synthesized by hydrothermal methods. The piezo/photodegradation rate (k value) of Rhodamine B (RhB) for BiOCl/NaNbO_3_ (BN-3, 0.0192 min^−1^) is 2.2 and 5.2 times higher than that of BiOCl (0.0089 min^−1^) and NaNbO_3_ (0.0037 min^−1^), respectively. The enhanced performance of BN-3 composite can be attributed to the heterojunction construction between BiOCl and NaNbO_3_. In addition, the piezo/photodecomposition ratio of RhB for BN-3 (87.4%) is 8.8 and 2.2 times higher than that of piezocatalysis (9.9%) and photocatalysis (40.4%), respectively. We further investigated the mechanism of piezocatalysis, photocatalysis, and their synergy effect of BN-3 composite. This study favors an in-depth understanding of piezo-photocatalysis, providing a new strategy to improve the environmental pollutant remediation efficiency of piezoelectric composites.

## 1. Introduction

There is increasing social concern over the current energy and environmental issues, especially wastewater pollution (originating from rapid industrial development), which poses a direct threat to human health [1,2]. Researchers are seeking effective, cost-efficient, stable, and safe ways to degrade and remove hazardous compounds in water. Semiconductor photocatalysis is one of the prominent strategies, not only converting sustainable solar energy into hydrogen energy but also utilizing visible light to degrade organic pollutants. A large number of semiconductor photocatalysts and their composites have been reported to improve the water splitting or pollutant degradation performance, such as ZnO and WO_3_ [3,4,5]. Among the various semiconductors, bismuth oxychloride (BiOCl) as a p-type bismuth oxyhalides semiconductor has an open layered structure composing of [Bi_2_O_2_]^2+^ layers sandwiched between [Cl_2_]^2−^ plates, which can facilitate the separation of photo-produced electrons and holes due to the more space to polarize the atoms and orbitals involved [6]. However, BiOCl is limited by its wide band gap (~3.4 eV), representing favorable photocatalytic performance only under ultraviolet light irradiation. Therefore, heterojunction–construction engineering is one of the most powerful strategies to achieve a broader photoresponse and improved photocatalytic activity by constructing heterostructured BiOCl photocatalysts [7,8,9].

In addition to photocatalysis, piezoelectric catalysis is also an efficient and environmentally friendly dye degradation method [10,11,12]. Vibration is one of the most common sources of energy in our environment. In piezoelectric catalysis, mechanical vibration generates an electric charge on the surface of the piezoelectric catalyst through the piezoelectric effect, which in turn reacts with the dye molecules, leading to the decomposition of the dye [13,14]. Combining with photocatalysis, piezoelectric materials possess the ability to reduce the carrier recombination rate due to the existence of an internal piezoelectric field in piezoelectric materials. In past decades, perovskite niobates, including NaNbO_3_, have attracted intensive attention because of their nonlinear optics, ionic conductive, piezoelectric, and photocatalytic properties [15,16,17]. In addition, the piezo-photocatalysis and pyroelectric catalysis of NaNbO_3_ have been thoroughly studied in recent years. For example, Jia’s group has found that NaNbO_3_ nanofibers possess a highly efficient piezoelectrically and pyroelectrically bi-catalysis for decomposition of organic dye [18]. It is worth noting that BiOCl is a piezoelectric material with piezocatalytic activity in response to ultrasound. Jia’s group have also reported that BiOCl shows highly efficient dye wastewater decomposition under the condition of light (300 W Xenon lamp) and ultrasound (120 W, 40 kHz) together, which is much greater than that of only with light or only with ultrasound, respectively [19]. However, there are not any reports for piezo-photocatalysis of BiOCl/NaNbO_3_ heterojunction piezoelectric composite. Therefore, it is necessary to explore the mechanism of BiOCl/NaNbO_3_ piezoelectric composite for enhancing degradation efficiency by using the synergistic effect of piezoelectric catalysis and photocatalysis.

In this work, the BiOCl/NaNbO_3_ piezoelectric composites were prepared by the hydrothermal method. The polarization electric field hysteresis loop (P-E) and electric-field-induced strain (S-E) curves confirm the BiOCl/NaNbO_3_ composite has good ferroelectric and piezoelectric properties. The catalytic performance of BiOCl/NaNbO_3_ piezoelectric composite was remarkably enhanced by the heterojunction construction and the synergy effect of piezocatalysis and photocatalysis, which greatly promote the separation of electron-hole pairs under electric field.

## 2. Materials and Methods

### 2.1. Materials Fabrication

#### 2.1.1. Synthesis of NaNbO_3_

The NaNbO_3_ powder was prepared by the hydrothermal method. Briefly, 1 g Nb_2_O_5_ was added into 30 mL sodium hydroxide (NaOH, 10 M) aqueous solution and stirred for 2 h. Then, transferred above solution into a 50 mL Teflon-lined autoclave and reacted at 180 °C for 48 h. The precipitate was washed thoroughly and dried at 80 °C.

#### 2.1.2. Synthesis of BiOCl

In brief, 5 mmol bismuth nitrate (Bi(NO_3_)_3_·5H_2_O, 99.5%) and 5 mmol potassium chloride (KCl) were added into 30 mL deionized water (DI water) and stirred for 0.5 h. Then, the above solution was transferred into a 50 mL Teflon-lined autoclave and heated at 180 °C for 12 h. The precipitate was washed thoroughly and dried at 80 °C.

#### 2.1.3. Synthesis of BiOCl/NaNbO_3_ Composites

Different contents (1, 2, 3, 4 mmol) of NaNbO_3_ (prepared in 2.1.1) were added into the 30 mL DI water and stirred for 0.5 h, and then, 5 mmol Bi(NO_3_)_3_·5H_2_O and 5 mmol KCl were added into the solution and stirred for another 0.5 h. The above solution was transferred into a 50 mL Teflon-lined autoclave and reacted at 180 °C for 12 h. The corresponding composites were named as BN-1, BN-2, BN-3, BN-4.

### 2.2. Photocatalytic Performance Experiment

The piezo/photocatalytic activities were evaluated by the degradation of RhB under UV–vis light irradiation and ultrasound. The 0.1 g catalyst was added into the RhB aqueous solution (100 mL, 5 mg/L) in dark and stirred for 30 min to reach adsorption–desorption equilibrium. After that, the mixed solution was treated with UV–vis light irradiation (300 W Xe lamp) and ultrasonic (50 W, 40 kHz). Change the water every five minutes to avoid the effects of temperature. A 4 mL solution was taken out every 20 min and centrifuged to remove the catalyst. The residual amount of RhB was recorded by the UV–vis spectrophotometer (Yoke, N6000, Shanghai, China) within the range of 300–800 nm.

### 2.3. Characterization

X-ray diffractometer (XRD) patterns were obtained to validate the phase purity and crystallinity of the powders on the XRD equipment (Rigaku Smartlab Beijing Co, Beijing, China). Scanning electron microscope (SEM) images of the prepared catalysts, including energy dispersive X-ray spectroscopy (EDS) capabilities, were measured with an SEM Regulus 8230, Hitachi Co, Tokyo, Japan. The transmission electron microscope (TEM) was used by JEOL JEM-F200 (Tokyo, Japan). The absorption spectra of these powders were tested in a UV-vis spectrophotometer (PerkinElmer Lambda 35, Waltham, MA, USA). X-ray photoelectron spectroscopy (XPS) was carried out with a ESCALAB 250, Thermo-VG Scientific, Waltham, MA, USA to analyze the components and the valence states. The specific surface areas of the samples were tested by Micromeritics ASAP 2460 Brunauer-Emmet-Teller (BET, Shanghai, China) equipment with N_2_ as the carrier gas. The polarization electric field (P-E) loops and electric-field-induced strain (S-E) were tested in silicone oil at room temperature with 1 Hz frequency using a MultiFerroic II, Radiant technologies Inc., Albuquerque, New Mexico. The sample powders were pressed in a pellet (1 cm diameter and 0.20 mm thick) with Polyvinyl Alcohol (PVA) solution as a binder and then annealed at 600 °C to burn out the PVA binder. The pellets were coated on both sides with Au electrodes.

## 3. Results and Discussion

X-ray diffraction patterns (XRD) of BiOCl, NaNbO_3,_ and a series of BiOCl/NaNbO_3_ piezoelectric composites are shown in Figure 1a. The distinct diffraction peaks of pure BiOCl can be related to tetragonal BiOCl (PDF card no. 82-0485, space group: P4/nmm), and the diffraction peaks of pure NaNbO_3_ can be indexed to orthorhombic NaNbO_3_ (PDF card no. 77-0873, space group: P2_1_ma). As for BiOCl/NaNbO_3_ piezoelectric composites (BN-1, BN-2, BN-3, BN-4), there are both BiOCl and NaNbO_3_ peaks can be observed. In addition, the crystallite sizes were calculated by Scherrer formula: $D=\frac{K\lambda }{\beta cos\theta }$, where *D* is crystallite size (nm), *K* is 0.9 (Scherrer constant), *λ* is 0.15406 nm (wavelength of the X-ray sources). The average crystallite sizes of BiOCl, NaNbO_3_, and BN-3 are 57, 22, and 56 nm. With the increase of NaNbO_3_ content, the diffraction peaks increased. The UV–vis diffuse reflectance spectra (DRS) of BiOCl, NaNbO_3,_ and BN-3 are exhibited in Figure 1b, which indicate the absorbance threshold of NaNbO_3_, BiOCl, and BN-3 are the same. The estimated band gaps (*E_g_*) of BiOCl and NaNbO_3_ are computed in Figure 1c by ${\left(Ahv\right)}^{2/n}\;\sim \;hv-E_{g}$, where *A* is for absorbance, *hv* is for irradiation energy [20], and the obtained values are 3.44 and 3.52 eV, respectively. The valance band X-ray photoelectron spectroscopy (VB XPS) spectra in Figure 1d show that the valance band values of BiOCl and NaNbO_3_ are 2.57 and 2.50 eV. Together with the band gaps, the conductive band (CB) position can be calculated by $E_{VB}=E_{CB}-E_{g}$, which are −0.87 (BiOCl) and −1.02 eV (NaNbO_3_).

The morphology and microstructure of the BN-3 powder were investigated by scanning electron microscopy (SEM), element mapping and transmission electron microscope (TEM), and the results were shown in Figure 2. From Figure 2a, the irregular particles can be observed and the distribution of the corresponding main elements are shown in Figure 2b–f. The different colored areas suggest that Nb-, Na-, O-, Bi-, and Cl-enriched areas of the BN-3 composite, respectively. The TEM image of the BN-3 powder is displayed in Figure 2g. The lattice spacing of 0.343 and 0.273 nm in Figure 2h–i are corresponding to the (101) of BiOCl and (121) plane of NaNbO_3_, respectively. The result agrees well with that in the XRD patterns as shown in Figure 1a. 

X-ray photoelectron spectroscopy (XPS) spectra of the BiOCl, NaNbO_3_, and BN-3 piezoelectric composite are shown in Figure 3. From Figure 3a, the peaks of Bi 4f of BiOCl (BN-3) located at 159.60 (159.24 eV) and 164.90 eV (164.52 eV) can be assigned to Bi 4f_7/2_ and Bi 4f_5/2_, respectively, suggesting the Bi^3+^ exists in the BiOCl (BN-3). In Figure 3b, the Cl 2p peaks at 198.29 (197.90 eV) and 199.93 eV (199.54 eV) can be attributed to Cl 2p_3/2_ and Cl 2p_1/2_, respectively, which indicate the Cl^−^ in BiOCl (BN-3) [21]. The peak at 1070.62 eV (1071.42 eV) in Figure 3c is ascribed to Na 1s in NaNbO_3_ (BN-3). In Figure 3d, it is clearly seen that the binding energies located at 206.67 (207.05 eV) and 209.40 eV (209.78 eV) belong to Nb 3d_5/2_ and Nb 3d_3/2_, respectively, reflecting that Nb is in the Nb (+5) chemical state [22]. As shown in Figure 3e, the peaks located at 530.39 (BiOCl), 529.60 (NaNbO_3_), and 529.98 eV (BN-3) correspond to O 1s. Compared with BiOCl, the blue shift of all peaks for BN-3 can be observed, while the red shift compared with NaNbO_3_. The XPS survey spectra also indicate that BiOCl is composed of Bi, O, and Cl elements, and NaNbO_3_ is mainly composed of Na, O, and Nb elements, while BN-3 contains all elements above, as shown in Figure 3f. In short, the XPS results demonstrate that the BN-3 piezoelectric composite is composed of BiOCl and NaNbO_3_.

The polarization electric field hysteresis loop (P-E) and electric-field-induced strain (S-E) curves of BN-3 composite are displayed in Figure 4. From Figure 4a, a saturated and nearly squared P-E loop can be observed, and the remnant polarization (Pr) is 35.13 μC/cm^2^ and the coercive field (Ec) is 8.72 kV/mm. The result shows that BN-3 composite has well ferroelectric properties, favoring the spatial separation and transportation of photo-induced carriers [23]. The S-E curve in Figure 4b exhibits an asymmetric butterfly shape, confirming the piezoelectricity of the BN-3 composite [24,25].

Consequently, the piezo/photocatalytic activities of BiOCl, NaNbO_3_, and BiOCl/NaNbO_3_ piezoelectric composites were evaluated by the degradation of Rhodamine B (RhB) under the condition of light irradiation and ultrasound. From Figure 5a, BN-3 exhibits better piezo/photocatalytic performance than that of BiOCl, NaNbO_3_, and other content BiOCl/NaNbO_3_ composites. The rate constant *k* values are obtained from Figure 5a via the pseudo-first-order equation [26]: $\ln\left(C_{0}/C_{t}\right)=-kt$, where *C*_0_ is RhB concentration for initial and *C_t_* is for after irradiation time *t*. And the decomposition ratio is calculated via the formula: $\eta =\left(1-\frac{C_{t}}{C_{0}}\right)\times 100\%.$ As shown in Figure 5b, the apparent reaction rate constant *k* for BiOCl, BN-1, BN-2, BN-3, BN-4, and NaNbO_3_ is 0.0089, 0.0112, 0.0134, 0.0192, 0.0168, and 0.0037 min^−1^, respectively. The piezo/photodegradation rate of RhB for BN-3 is 2.2 and 5.2 times higher than that of BiOCl and NaNbO_3_, the histogram in Figure 5c reflects this directly. The degradation percentages of BiOCl, NaNbO_3_ and BN-3 are 29.8%, 61.9% and 87.4%, respectively. In addition, the BET surface areas of BiOCl, NaNbO_3,_ and BN-3 are 0.24, 2.00, and 1.32 m^2^/g, and the pore volumes are 0.0008, 0.0088, and 0.0037 cm^3^/g, respectively. The BET surface areas of the samples are in the same order of magnitude, which means the BET surface areas not can decisive the catalytic activity. This result indicates the heterojunction in BN-3 exerts a tremendous advantage on the piezo/photocatalytic process. To investigate the most reactive species during the process of RhB decomposition, the radical trapping experiments were carried out in the presence of BN-3 as a catalyst. From Figure 5d, the piezo-photodegradation efficiency of RhB is remarkably inhibited while adding the triethanolamine (TEOA, 50 µL) scavenger for trapping hole (h^+^) to the mixed solution, demonstrating an important role of h^+^ in the piezo-photocatalytic process. While L-ascorbic acid (VC, 40 mg) for superoxide radical (·O_2_^−^) was added, the degradation efficiency also decreased rapidly. The RhB degradation efficiency is decreased slightly by adding the isopropanol (IPA, 50 µL), reflecting the hydroxyl radical (·OH) plays a secondary role in this process. These results indicate that the effect in this piezo-photocatalytic process is: h^+^ >·O_2_^−^ >·OH. 

To demonstrate the piezocatalysis, photocatalysis, and the synergy effect of piezocatalysis and photocatalysis of BN-3 piezoelectric composite, the RhB degradation capability within 100 min was measured under the condition of ultrasound only, light only, and ultrasound + light together. UV–vis absorption spectra of RhB for BN-3 under different conditions are shown in Figure 6a,c,e, which correspond to light, ultrasound, and both light and ultrasound, respectively. From Figure 6b,d,f, the degradation rate is the lowest under the condition of only ultrasound, the rate constant *k* (0.0005 min^−1^) and decomposition ratio (9.9%) are well below those of the condition of only light (0.0044 min^−1^ and 40.4%) and light + ultrasound together (0.0192 min^−1^ and 87.4%). The decomposition ratio of RhB under synergy of piezocatalysis and photocatalysis is 8.8 and 2.2 times higher than that of piezocatalysis and photocatalysis, respectively. The rate constant *k* under synergy of piezocatalysis and photocatalysis is 38.4 and 4.36 times higher than that of only ultrasound and only light. In addition, based on the same condition, compared to other piezoelectric materials past reported, the k value of BN-3 is higher than that of NaNbO_3_/CuBi_2_O_4_ nanocomposites (0.0112 min^−1^) [27], and closing to that of BaTiO_3_/KNbO_3_ heterostructure (0.01492 min^−^^1^) [28]. The result confirms that the synergy effect of piezocatalysis and photocatalysis of BiOCl/NaNbO_3_ piezoelectric composite plays an important role in the highly efficient degradation of RhB. One of the key parameters in the piezo-photocatalyst is reproducibility, and Figure 7 shows the cycling performance of the piezo-photocatalytic activity of BN-3 for degrading RhB. After three cycles, the degradation efficiency is just reduced a little. This result evidences that BN-3 possesses a high reproducibility.

On the basis of the above analysis, the possible mechanism for piezocatalytic, photocatalytic, and their synergetic catalytic process of BiOCl/NaNbO_3_ piezoelectric composites are shown in Figure 8. According to our experiment, the valance band of NaNbO_3_ is 2.50 eV, while BiOCl is 2.57 eV; the conductive band of NaNbO_3_ is −1.02 eV, while BiOCl is −0.87 eV. In the condition of only light, the photoelectrons are excited from the valance band to the conductive band, and the electrons will transfer from the conductive band of NaNbO_3_ to the conductive band of BiOCl, and thus build an inner electric field. The built-in electric field can promote the separation of electrons and holes. However, there is still a combination of electrons and holes in the inner of BiOCl/NaNbO_3_ piezoelectric composites because the built-in electric field is easily prone to be screened by electrostatic compensated free space charges [29]. This reduces the degradation efficiency of RhB. In the condition of only ultrasound, the cavitation bubbles will form, expand, and burst, an amount of electric charge can be generated [30,31]. These positive and negative charges will transfer to the opposite directions under the influence of the alternating built-in electric field. In the condition of both light and ultrasound, the electrons and holes located at the conductive band and valance band will transfer to the opposite directions under the internal piezoelectric potential, causing electrons to accumulate in the conductive band of BiOCl and holes accumulate in the valance band of NaNbO_3_ [32,33,34]. Subsequently, the electrons on the CB of BiOCl combined with the absorbed O_2_ to produce ·O_2_^−^. Meanwhile, part holes on the VB of NaNbO_3_ will oxidize hydroxyl to form ·OH. Finally, the reactive species ·OH, h^+^, and ·O_2_^−^ will participate in the oxidative degradation of RhB. The combination rate of electrons and holes will be reduced significantly under the built-in electric field, thus the decomposition ratio of BiOCl/NaNbO_3_ piezoelectric composite increased remarkably.

## 4. Conclusions

In conclusion, BiOCl/NaNbO_3_ piezoelectric composites are synthesized via a two-step hydrothermal route. Under UV–vis light and ultrasonic exposure, the BN-3 for piezo-photocatalytic decomposition of RhB demonstrate remarkable piezo-photocatalytic performance than that of BiOCl and NaNbO_3_ component due to the heterojunction construction. Furthermore, the piezo/photodegradation rate of RhB for BN-3 is higher than that of piezocatalysis and photocatalysis, indicating the synergistic effect of piezocatalysis and photocatalysis plays a significant role in the degradation process. Some issues, such as the specific promotion mechanism of the NaNbO_3_ and BiOCl of the contribution of the piezo-photocatalytic performance, need to be further investigated for better understanding. However, this work provides a feasible approach for the development of efficient piezoelectric photocatalysts for heterojunction construction.

## Figures and Tables

**Figure 1 nanomaterials-12-00353-f001:**
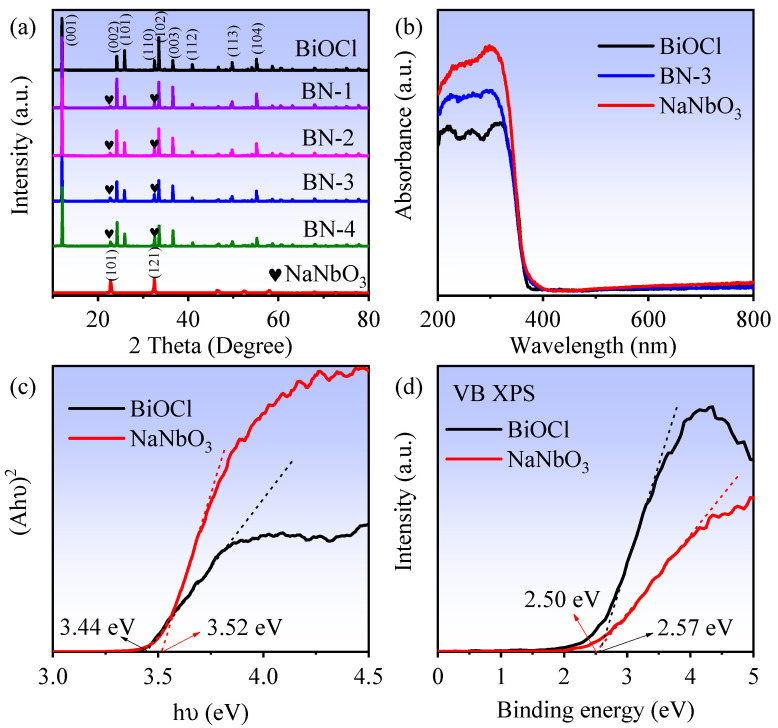
(**a**) XRD patterns of BiOCl, NaNbO_3_, and BiOCl/NaNbO_3_ piezoelectric composites; (**b**) UV–vis absorption spectra; (**c**) the estimated band gaps of BiOCl and NaNbO_3_; (**d**) VB XPS spectra of BiOCl and NaNbO_3_.

**Figure 2 nanomaterials-12-00353-f002:**
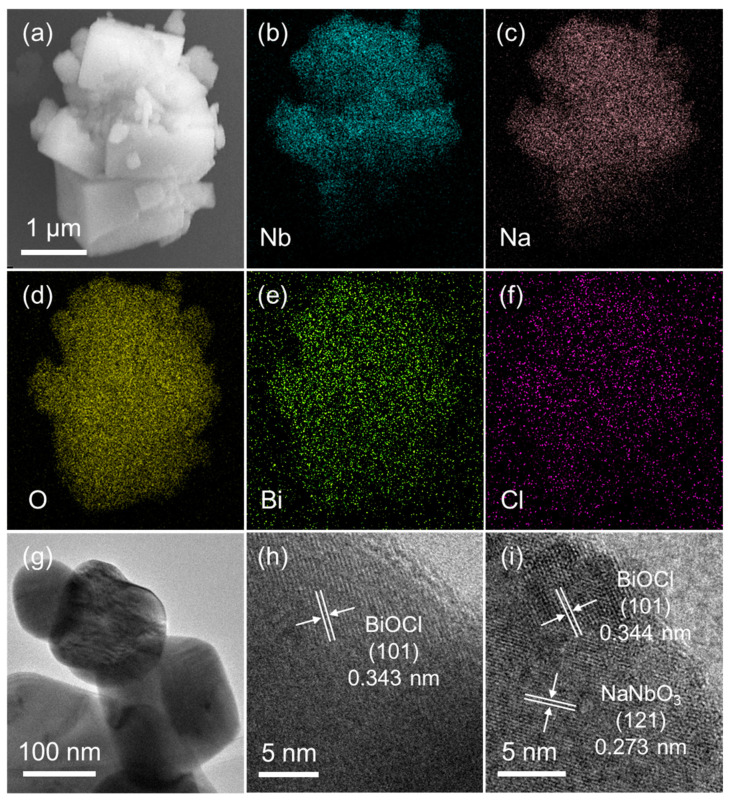
(**a**) SEM image; (**b**–**f**) EDS element mappings; (**g**) TEM image; (**h**,**i**) lattice fringes images of BN-3.

**Figure 3 nanomaterials-12-00353-f003:**
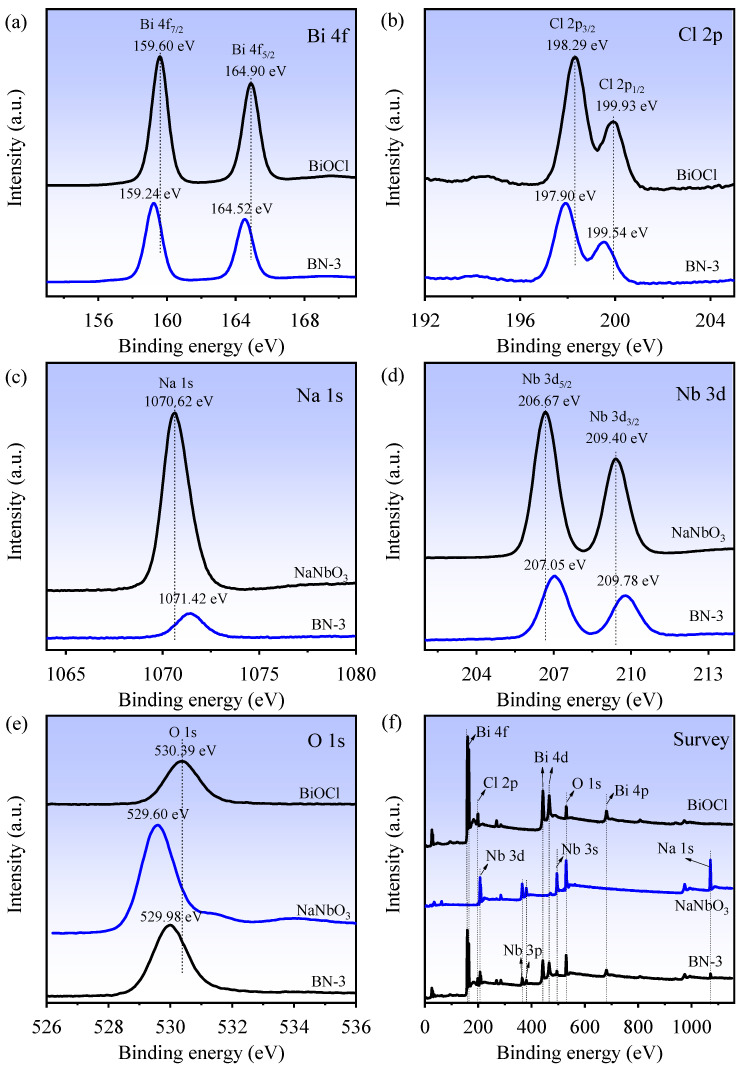
XPS survey spectra of BiOCl, NaNbO_3_, and BN-3: (**a**) Bi 4f; (**b**) Cl 2p; (**c**) Na 1s; (**d**) Nb 3d; (**e**) O 1s; (**f**) Survey.

**Figure 4 nanomaterials-12-00353-f004:**
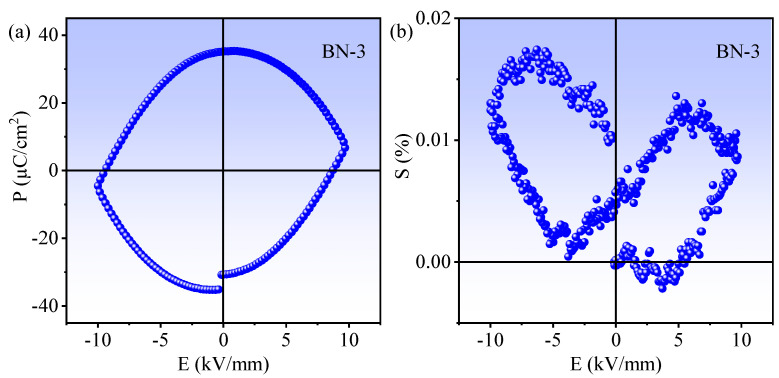
(**a**) The ferroelectric P-E loop and (**b**) electric field-induced S-E curve of BN-3 composite.

**Figure 5 nanomaterials-12-00353-f005:**
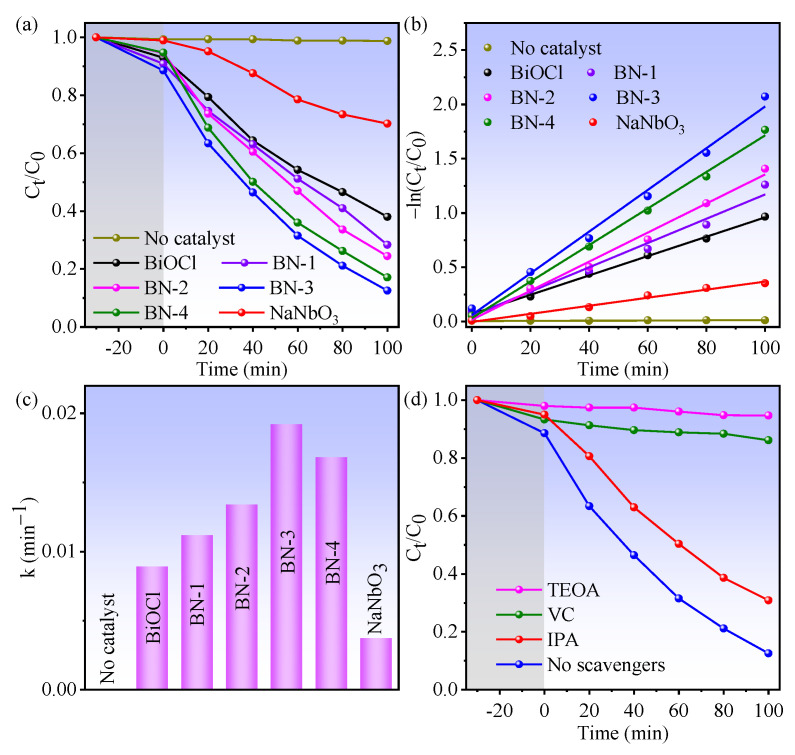
(**a**) The kinetic curves of piezo-photodegradation RhB performance for BiOCl, NaNbO_3_, and BiOCl/NaNbO_3_ piezoelectric composites; (**b**) the dynamics of degradation reaction [(−ln(*C_t_*/*C*_0_)]; (**c**) the histogram of corresponding reaction rate constant; (**d**) piezo-photodegradation curves with disparate scavengers of BN-3 composite.

**Figure 6 nanomaterials-12-00353-f006:**
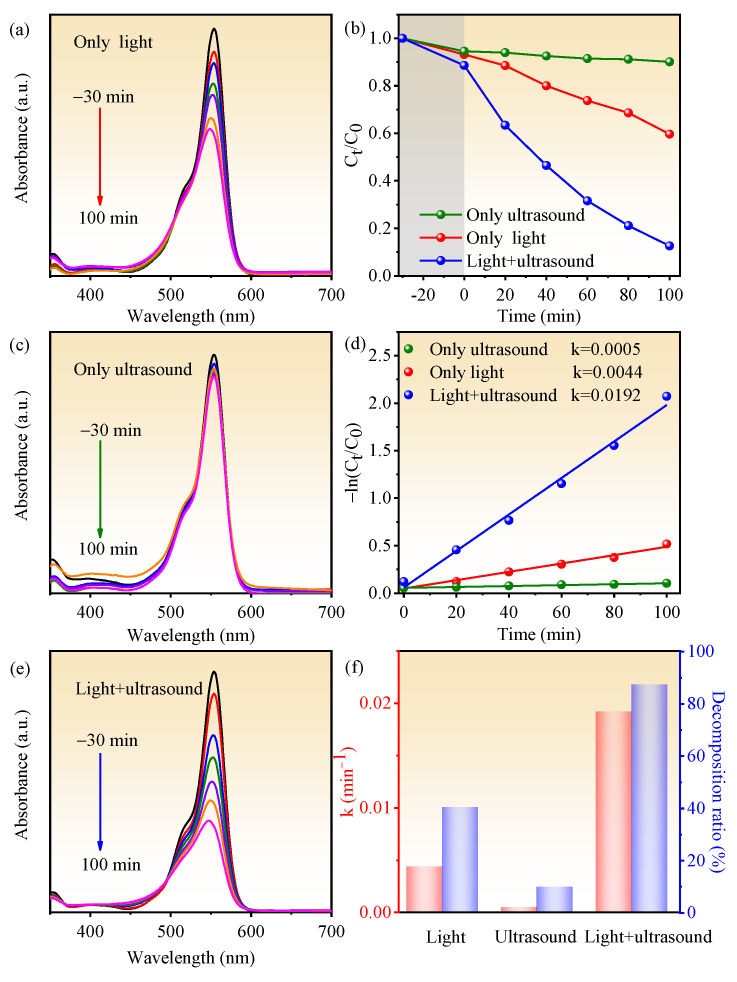
UV–vis spectral absorption of RhB for BN-3 under the condition of (**a**) only light, (**c**) only ultrasound and (**e**) light + ultrasound; (**b**) the kinetic curves of RhB degradation for BN-3 under these three control conditions; (**d**) the dynamics of degradation reaction [(−ln(C_t_/C_0_)]; (**f**) the histogram of corresponding reaction rate constant and decomposition ratio.

**Figure 7 nanomaterials-12-00353-f007:**
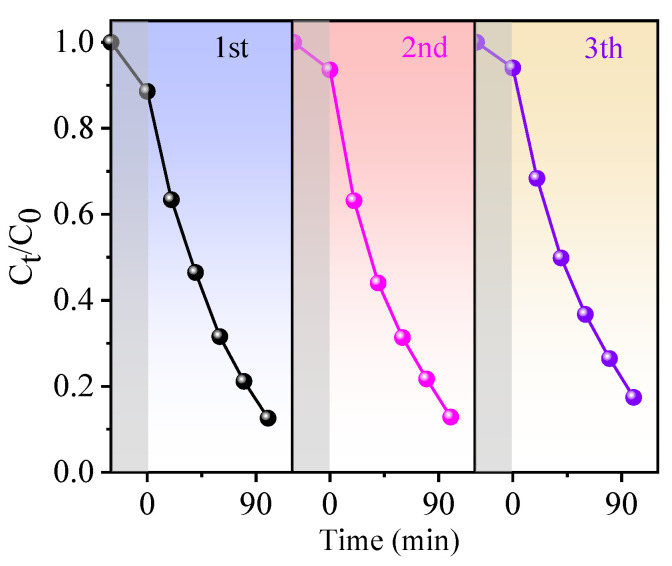
The cycling performance of the piezo-photocatalytic activity of BN-3 for degrading RhB solution.

**Figure 8 nanomaterials-12-00353-f008:**
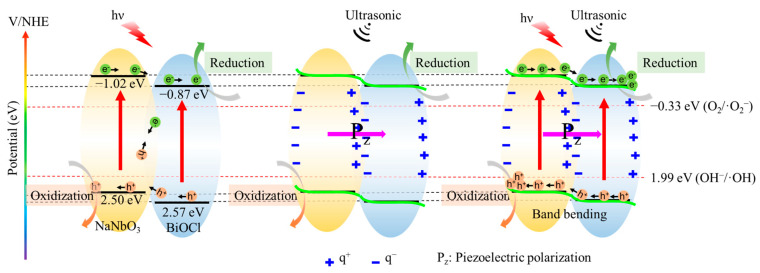
Possible piezocatalytic, photocatalytic, and piezo-photocatalytic mechanism of BiOCl/NaNbO_3_ piezoelectric composites.

## Data Availability

Data available in a publicly accessible repository. The data presented in this study are openly available in [repository name e.g., FigShare] at [doi], reference number [reference number].

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
