# Peer review of "Synergistic Piezo-Photocatalysis of BiOCl/NaNbO3 Heterojunction Piezoelectric Composite for High-Efficient Organic Pollutant Degradation"

_nanomaterials, 2022, doi:10.3390/nano12030353_

Round 1
Reviewer 1 Report
Comments to Author
In this manuscript, the author focused on the removal of Rhodamine B with BiOCl/NaNbO3 heterojunction piezoelectric composite by using the piezo-photocatalysis technique. There is no novelty in your work and so many reports are available on the pollutant Rhodamine B. Hence, I recommended a major review and it is accepted for this journal after the author clarifies the following comments.
- The Author should correct the title it is “organic pollutant degradation”, not “organic pollution degradation.
- The Author should enhance the novelty and importance of this work in the introduction section.
- In the experimental section, the synthesis method of composite BiOCl/NaNbO3 was not clear. The author should rewrite it.
- The Author should include the crystalline planes related to BiOCl and NaNbO3 in the XRD figure (Fig. 1a) and the discussion part related to XRD.
- What are the crystallite sizes of BiOCl, NaNbO3, and BiOCl/NaNbO3 composites?
- The Y-axis title in Fig. 1(b) and titles of X-axis and Y-axis in Fig. 1(c) are wrong. The Author should correct it. What is the bandgap energy of the BN-3 composite?
- The SEM and TEM images are unclear. The Author should increase the resolution. The legends of Fig. 2 (h & i) are wrong these are lattice fringes and not HRTEM. The Author should correct it.
- In photocatalytic degradation surface area plays an important role, the author should calculate the BET-specific surface areas of all catalysts that are used in this work?
- Which light source was used for the photocatalysis method and what is the intensity of the light source?
- What are the reaction conditions (Catalyst amount, Concentration of Rhodamine B, pH) for the piezo-photocatalysis technique?
- What are the degradation percentages of BiOCl, NaNbO3, and BiOCl/NaNbO3 composites?
- The Author should comment on the mineralization of pollutants in the solution after the reaction. How to calculate Rhodamine B mineralization percentage after the reaction and any analytical technique is there?
- The manuscript contains many typos and the language should be polished.
Reviewer 2 Report
This work aimed to synthesized Bi-OCl/NaNbO3 with different molar ratios piezoelectric composites by hydrothermal methods. The piezo/photodegradation rate of Rhodamine B (RhB) for BiOCl/NaNbO3 (BN–3) is 2.2 and 5.2 times higher than that of BiOCl and NaNbO3, respectively. The enhanced per-formance of BN–3 composite can be attributed to the heterojunction construction between BiOCl and NaNbO3. In addition, the piezo/photodecomposition ratio of RhB for BN–3 is 8.8 and 2.2 times higher than that of piezocatalysis and photocatalysis, respectively. However, some critical issues remain to be solved and a thorough revision was needed:
- The introduction should be clarified in terms of uniqueness and advantage what is the novelty of this work over the previous related work, and cite the following references:
J. Mol. Struct. 1191 (2019) 76-84;
J. Chin. Chem. Soc. 66 (2019) 89-98;
- On what basis the authors chose this particular dye. Did they check for other dye also?
- What about the pore volume and surface area of the adsorbent? BET analysis is important
- What about the kinetic study of degradation?
- Regeneration and stability of the photocatalyst?
- The authors should make comparison with literature for the dye used.
- The manuscript needs thorough revision to improve the text quality and readability of the work.
Reviewer 3 Report
The authors reported the development of BiOCl/NaNbO3 piezoelectric composites prepared by the hydrothermal method. The characterization aspect of the prepared composites was sufficient, and the practical aspect (piezocatalysis and photocatalysis) was reasonably and sufficiently studied. This manuscript needs major revision and I have listed these issues and recommendations in chronological order. Following is a summary of the major corrections and revisions:
- The authors may need to briefly address the difference(s) between the current manuscript and other similar published review articles in the Introduction section.
- In general, the references in the introduction are poorly chosen, when compared to the sentences they serve as confirmation for.
- Please explain why you have used this specific combination of BiOCl with NaNbO3, instead of other perovskite niobates. What is the advantage?
- In section 2, the details regarding the experimental part of the piezo–photodegradation RhB performance e for BiOCl, NaNbO3, and BiOCl/NaNbO3 piezoelectric composites are missing. Please add details regarding the experimental aspects. Also, details regarding the UV–vis spectral absorption of RhB for BN–3 are not in the manuscript.
- Conclusions: Conclusions need to be improved by specifying the discussed important points within this work. In the conclusions, the authors should also provide an outlook of the challenges and potential future directions.
Other comments:
- Overall, the materials used should be more characterized. For instance, I suggest the authors to measure zeta potential values of the materials as function of pH and to measure the specific surface area.
- Did the authors check any interferences in spectrophotometer measurements of dye from the matrix of the synthesized materials? Please comment.
- Could you comment on whether there is aging of the samples during time? This is absolutely important for practical applications.
Round 2
Reviewer 1 Report
Comments to Author
In this manuscript, the author focused on the removal of Rhodamine B with BiOCl/NaNbO3 heterojunction piezoelectric composite by using the piezo-photocatalysis technique. Hence, I recommended a minor review and it is accepted for this journal after the author clarifies the following comments.
- “The synthesis method was the same as BiOCl, except for NaNbO3 During the process of the synthesis of BiOCl, different contents (1, 2, 3, 4 mmol) of NaNbO3 were added into the solution and reacted at 180 °C for 12 h.” This paragraph is unclear for the synthesis of composite. The Author should correct it.
- Calculate the crystallite sizes of BiOCl, NaNbO3, and BiOCl/NaNbO3 composites by Debye- Scherer formula?
- The bandgap energy of the BiOCl/NaNbO3 composite is 3.48 eV. It shows activity in the UV region not in the visible region. But in your work, you used a visible light source for photocatalytic reactions. How does the composite show activity in the visible region?
- In the legends of Figure 2, only 2 (g) is the TEM image, and 2(h) and 2(i) are a representation of lattice fringes. The Author should correct it.
- How much mineralization was done for composite after the photocatalytic reaction? How to measure mineralization percentage for Rhodamine B after analysis?
Reviewer 3 Report
The authors have addressed my queries satisfactorily. Manuscript was improved in accordance to my suggestions and I have no further objection to this paper. The quality of the paper has been improved and, therefore, I consider that the article can be accepted in present form.
Author Response
Thank you very much.